# High-Throughput Sequencing Reveals Transcriptome Signature of Early Liver Development in Goat Kids

**DOI:** 10.3390/genes13050833

**Published:** 2022-05-06

**Authors:** Xiaodong Zhao, Rong Xuan, Aili Wang, Qing Li, Yilin Zhao, Shanfeng Du, Qingling Duan, Yanyan Wang, Zhibin Ji, Yanfei Guo, Jianmin Wang, Tianle Chao

**Affiliations:** 1Shandong Provincial Key Laboratory of Animal Biotechnology and Disease Control and Prevention, College of Animal Science and Veterinary Medicine, Shandong Agricultural University, Taian 261018, China; zxd18763895097@163.com (X.Z.); 2018110375@sdau.edu.cn (R.X.); 18853852560@163.com (Q.L.); z18853811161@163.com (Y.Z.); dushanfeng1998@163.com (S.D.); duanql923@163.com (Q.D.); 17826621820@163.com (Y.W.); zbji916@sdau.edu.cn (Z.J.); guoyf416@163.com (Y.G.); wangjm@sdau.edu.cn (J.W.); 2Center for Evolution and Conservation Biology, Southern Marine Science and Engineering Guangdong Laboratory (Guangzhou), Guangzhou 511458, China; allie612@163.com

**Keywords:** goat kid, liver development, transcriptome, regulatory network

## Abstract

As a vital metabolic and immune organ in animals, the liver plays an important role in protein synthesis, detoxification, metabolism, and immune defense. The primary research purpose of this study was to reveal the effect of breast-feeding, weaning transition, and weaning on the gene expression profile in the goat kid liver and to elucidate the transcriptome-level signatures associated with liver metabolic adaptation. Therefore, transcriptome sequencing was performed on liver tissues, which was collected at 1 day (D1), 2 weeks (W2), 4 weeks (W4), 8 weeks (W8), and 12 weeks (W12) after birth in Laiwu black goats at five different time-points, with five goats at each time point. From 25 libraries, a total of 37497 mRNAs were found to be expressed in goat kid livers, and 1271 genes were differentially expressed between at least two of the five time points. Gene ontology (GO) and Kyoto Encyclopedia of Genes and Genomes (KEGG) pathway analyses revealed that these genes were annotated as an extracellular region fraction, exhibiting monooxygenase activity, positive regulation of T cell activation, mitotic spindle mid-region assembly, cytokinesis, cytoskeleton-dependent cytokinesis, regulation of cytokinesis, regulation of lymphocyte proliferation, and so on. In addition, it mainly deals with metabolism, endocrine, cell proliferation and apoptosis, and immune processes. Finally, a gene regulatory network was constructed, and a total of 14 key genes were screened, which were mainly enriched for cell growth and development, endocrine, immune, and signal transduction-related pathways. Our results provide new information on the molecular mechanisms and pathways involved in liver development, metabolism, and immunity of goats.

## 1. Introduction

The liver is a central organ that plays a key role in nutrient metabolism, including the metabolism of proteins, fats, and carbohydrates [1,2,3]. Furthermore, liver also plays a crucial role in immunity, mainly due to the fact that the blood contains large amounts of gastrointestinal antigens circulating in the hepatic vasculature, helping to form an immune barrier against pathogens [4].

During embryonic and neonatal periods, the liver functions primarily as a key hematopoietic organ [5]. During postnatal development, this organ transitions to mature metabolic function, a process that dramatically changes liver gene expression and promotes a shift from a hepatic microenvironment to one that is extensively involved in metabolism and detoxification [6,7,8,9]. At present, most studies on the molecular mechanisms related to the early development and functional transformation of the liver mainly focus on the fetal stage, while the research on the neonatal and postnatal stages are relatively rare and need to be further studied.

During newborn developmental stages, such as breast feeding and weaning transition, the liver undergoes rapid physiological and genetic transformation in response to changes in nutrient intake [10,11,12]. In nutrient metabolism, the neonatal liver undergoes complex changes related to glucose and fatty acid metabolism after birth [13]. Mammalian neonates typically experience a dramatic drop in blood glucose levels immediately after birth and rise later, along with a decrease in plasma insulin levels and an increase in glucagon levels [14,15,16]. This process promotes the mobilization and consumption of glycogen in liver tissue [13]. After that, with the increase of phosphoenolpyruvate carboxykinase activity, the gluconeogenesis function of the liver is continuously enhanced until it provides about two-thirds of the glucose source [17]. At the same time, because mammalian milk is usually rich in lipids and relatively low in carbohydrates, fatty acid metabolism is an important energy source for neonates in the early postpartum period [13,18]. In the first day after birth, neonatal hepatic ketosis increases rapidly and makes ketone bodies one of the main energy sources for neonates [17,19]. The maturation of bile secretion and enterohepatic circulation is also an important part of neonatal liver development. During the late gestation, the activity of enzymes involved in bile acid synthesis increases, leading to an increase in the size of the bile acid pool before birth [20,21]. At birth, in-creased bile acid transport across the basolateral and canalicular membranes of hepato-cytes results in the transfer of bile acid pools from the liver to the gut, and sodium-bile acid co-transport activity in the terminal ileum is generally fully developed at weaning [22,23]. In addition, under the influence of subsequent changes in nutrient intake, weaning, and other processes, the physiological functions of neonates liver also undergoes corresponding complex adjustments and adaptations. However, most studies on neonatal liver development have been performed on species other than goats, while little is known about the changes and regulatory mechanisms in the goat liver during this process. At present, some studies have reported on the transcriptome level regulation of goat fetal or kid liver, but the study period focused on prenatal or post-weaning [24,25,26], and there is no report on postnatal goats during breast feeding, weaning transition, and weaning stages.

The objective of this study was to reveal the characteristics of the liver transcriptome of postnatal goat kids and to identify genes whose expression is affected by starter feeding and weaning. Therefore, in this study, goat liver samples were subjected to transcriptional analysis at 1 day and at 2, 4, 8, and 12 weeks to examine the differentially expressed genes (DEGs).

## 2. Materials and Methods

### 2.1. Sample Collection

The experimental animals used in this study were Laiwu black goat, which is a local goat breed with both cashmere and meat produced in the central mountainous area of Shandong Province. It has germplasm characteristics, such as high reproduction rate, excellent meat quality, and strong disease resistance [27]. All goat kids were obtained from Shandong Fengxiang Livestock Breeding Technology Co. Goat kids lived with their dams and suckled from their mother as the only food source for the first 30 days, which included the colostrum stage (D1), the middle nursing milk stage (W2), and the late nursing t milk stage (W4). Goats were supplemented with solid concentrate starting at 30 days, and fermentation of the solid concentrate was the only food source for the goat kid after 60 days. Based on this feeding strategy, we defined age after solid concentrate supplementation as the mixed-feeding stage (W8) and the starter-feeding stage (W12). All goat kids had ad libitum access to water and concentrate. On day 1, five newborn goat kids were sacrificed 6 h after suckling colostrum, and all other groups of goat kids were sacrificed at their respective cut-off dates to collect liver tissues (five goat kids in each time point). Overall, a total of 25 healthy goats (third parity) were selected as experimental animals. The collected liver samples were snap-frozen in liquid nitrogen.

### 2.2. RNA Extraction

The total RNA was extracted from the liver tissues with Trizol reagent, and the genomic DNA was removed by RNase-free DNase I. The Bioanalyzer 2100 (Agilent Technologies, Santa Clara, CA, USA) and the ND-2000 (NanoDrop Technologies, Wilmington, DE, USA) were used to examine the quality of RNA. The sequencing libraries were constructed using only high-quality RNA samples (OD260/280 = 1.8–2.2, from 260/230 ≥ 2.0, RIN ≥ 8, 28S: 18S ≥ 1.0, >10 µg).

### 2.3. Library Preparation and Sequencing

RNA-seq transcriptome strand libraries were set up from 5 ug of total RNA using the Illumina TruSeq TM Total Stranded RNA Library Preparation Kit (San Diego, CA, USA). The paired-end transcriptome libraries were sequenced by Shanghai Majorbio Bio-Pharm Biotechnology Co., Ltd. (Shanghai, China) using the Illumina NovaSeq6000 system.

### 2.4. Read Mapping and Transcriptome Assembly

SeqPrep (https://github.com/jstjohn/SeqPrep, accessed on 18 January 2021) and Sickle (https://github.com/najoshi/sickle, accessed on 18 January 2021) were used as default parameters for cropping and the quality control of raw-paired-end reads. The clean reads were then aligned with the goat reference genome (Assembly: GCA_001704415.1) in the direction mode using the HISAT2 software [28]. The associated reads for each example were assembled by the StringTie [29] in a reference-based approach (Goat reference genome annotation: Capra hircus ARS1). The expression level of each transcript was calculated with the fragment-per-kilobase-per-million-reads (FPKM) method.

### 2.5. Differential Expression Analysis and Functional Enrichment

Essentially, DESeq2 [30] was performed for the differential expression analysis, and differential expression genes (DEGs) with fold change >2 or <−2 and a Q value ≤ 0.05 were considered to be significantly differentially expressed genes. In addition, the mean FPKM values for at least one group were not <1.

The K-means (k = 4) clustering of the relationship between samples and transcripts was performed based on the expression levels of the transcripts, and the clustering results were presented in heatmaps. The expression scales of the 25 groups of samples were normalized by the Z-Score, and the corresponding heatmaps were constructed using MEV 4.9.0 [31].

In addition, we performed a functional enrichment analysis, including GO and KEGG, to determine which DEGs were significantly enriched in the GO terms and the pathways when compared to the transcriptome-wide background at a Bonferroni-corrected *p*-value ≤ 0.05. Goatools [32] and KOBAS [33] were used to perform the GO function enrichment and KEGG pathway analysis.

### 2.6. Construction of the Protein–Protein Interaction (PPI) Network

Across the 25 libraries, in accordance with the KEGG results, four clusters of DEGs were significantly enriched in 34 pathways containing a total of 148 genes. All 148 genes were used for the PPI network construction. Their interactions were predicted after in the database of STRING [34]. The protein-coding genes regulatory network was constructed using the Cytoscape [35].

### 2.7. Quantitative Real-Time PCR (qRT-PCR) Validation

A total of 15 DEGs were selected randomly for RT-qPCR analysis. The primers for controls were designed based on the sequences of the selected DEGs. The primers used in this study are shown in Table 1. cDNA was synthesized following the instructions of the Evo M-MLV RT Kit with gDNA Clean for qPCR II (AG11711, ACCURATE BIOTECHNOLOGY (Hunan) Co., Ltd., Changsha, China). RT-qPCR was performed using the SYBR Green Premix Pro Taq HS qPCR Kit (AG11701, ACCURATE BIOTECHNOLOGY (Hunan) Co., Ltd., Changsha, China) and the Roche LightCycler^®^ 96 System (Roche, Pleasanton, CA, USA). There were reaction volumes of 20 µL, 0.2 µL of forwarding primer, 0.2 µL of reverse primer, 10 µL of 2 × SYBR Green Pro Taq HS Premix, 2 µL of cDNA, and 7.6 µL of the RNase free water. The reaction conditions of qPCR are divided into two steps, the first step is 95 °C for 30 s, and the second step is 95 °C for 5 s, then 60 °C for 30 s, and this cycle was repeated 40 times. All reactions were performed in triplicate. β-actin (*ACTB*) was used as an internal reference gene. After amplification, the gene expression was calculated using the 2^−ΔΔCT^ ± SEM [36]. Correction for multiple tests were determined with one-way ANOVA, least-significant difference (LSD), and Tukey’s multiple range tests.

## 3. Results

### 3.1. Sequencing Data Summary

Twenty-five liver cDNA libraries encompassing five stages of the Laiwu Black goats were constructed. A total of 3,106,609,276 raw reads were obtained. After screening, 3,023,865,598 clean reads were obtained (Appendix A). Twenty-five libraries were compared with the goat genome at a rate of >96%. Specifically, D1-1, D1-2, D1-3, D1-4, D1-5, W2-1, W2-2, W4-2, W4-3, W4-4, W4-5, W8-1, W8-2, W8-3, W8-4, W8-5, W12-1, W12-2, W12-3, W12-4, and W12-5 were mapped at 97.45%, 97.19%, 97.18%, 97.42%, 97.40%, 96.94%, 97.30%, 97.38%, 97.10%, 97.21%, 97.09%, 97.26%, 97.16%, 97.19%, 97.39%, 97.23%, 97.52%, 97 97.51%, 97.30%, 97.42%, 97.32%, 96.99%, 97.51%, 97.40%, and 97.38%, respectively (Appendix A).

Appendix A displays the number of known and novel mRNAs in each library. As shown in Table 2, a total of 37,497 mRNAs were found in all libraries: 31,185 in D1, 31,369 in W2, 31,216 in W4, 31,064 in W8, and 31,143 in W12. Overall, 29,976 known mRNAs were detected in all libraries, which accounted for 79.94% of the total mRNAs. In the D1, W2, W4, W8, and W12 groups, 23,861 (73.37% of the total), 24,022 (76.58% of the total), 23,901 (76.57% of the total), 23,725 (76.57% of the total), and 23,725 (76.57% of the total) known mRNAs, respectively, were detected. A total of 7521 novel mRNAs were found in all libraries, including 7324 in D1, 7347 in W2, 7315 in W4, 7339 in W8, and 7344 in W12. The W8-1 library had the lowest number of known mRNAs, novel mRNAs, and total mRNAs (18,778, 7029, and 25,807, respectively). On the other hand, the D1-2 library had the most widely known mRNAs, novel mRNAs, and total mRNAs (19,620, 7091, and 26,711, respectively).

### 3.2. Identification and Analysis of DEGs

As listed in Appendix A and Figure 1, 1271 differential genes were used in the analyses. Cluster plots of differential expression patterns between the groups revealed that 1271 DEGs were categorized into four clusters: 424 DEGs in the first cluster, 248 DEGs in the second cluster, 217 DEGs in the third cluster, and 382 DEGs in the fourth cluster. The first cluster showed several DEGs between the D1 and other groups, and their expression levels decreased with age in Laiwu black goat (Figure 1A and Figure 2A). The second cluster showed a high expression of DEGs in the W2 and W8 groups and a low expression in the other groups (Figure 1B and Figure 2B). The third cluster showed a high expression of DEGs in the D1 and W4 groups and an upregulation of the low expression in the other groups (Figure 1C and Figure 2C). The fourth cluster showed several DEGs between the W12 and other groups, and their expression levels increased with age in the Laiwu black goat (Figure 1D and Figure 2D). The fourth clusters of DEGs were analyzed by GO and KEGG enrichment analysis.

### 3.3. GO and KEGG Enrichment Analysis

GO and KEGG enrichment analyses were performed on DEGs to further confirm their involvement in the regulation of biological processes. The results of the GO analysis are depicted in Appendix A. During the enrichment analysis of the first cluster of DEGs, 711 terms were obtained, all of which were not significantly enriched. In the second cluster of DEGs, 804 terms were obtained, all of which were not significantly enriched. In the enrichment analysis of the third cluster of DEGs, 690 terms were obtained, of which 34 terms were significantly enriched, including 19, 0, and 15 terms for the cellular component (CC), molecular function (MF), and biological process (BP) categories, respectively (FDR < 0.05). Microtubule (GO:0005874) and mitotic spindle midzone assembly (GO:0051256) revealed the highest enrichment significance in the CC and BP terms, respectively. In the enrichment analysis of the fourth cluster of DEGs, 1006 GO terms were obtained, of which 25 GO terms were significantly enriched, with 1, 1, and 23 GO terms in the CC, molecular function (MF), and BP terms, respectively (FDR < 0.05). The extracellular region (GO:0044421), monooxygenase activity (GO:0004497), and positive regulation of T-cell activation (GO:0050870) demonstrated the highest enrichment significance in the CC, MF, and BP terms, respectively. Notably, the top five significantly enriched terms for the third and fourth categories of DEGs belonged to BPs, with the top five significantly enriched entries for the third category associated with cell growth and development, which included mitotic spindle midzone assembly (GO:0051256, FDR = 0.00111856325616), cytokinesis (GO:0000910, FDR = 0.00187061126159), cytoskeleton-dependent cytokinesis (GO:0061640, FDR = 0.00384756153292), regulation of cytokinesis (GO:0032465, FDR = 0.00449815175238), and regulation of cell division (GO:0051302, FDR = 0.00599432465923); the first five significantly enriched entries in the fourth cluster were all immune-related, and these included the positive regulation of T-cell activation (GO:0050870, FDR = 0.00455494982368), regulation of lymphocyte proliferation (GO:0050670, FDR = 0.00555884532439), regulation of mononuclear cell proliferation (GO:0032944, FDR = 0.00596871231672), positive regulation of homotypic cell-cell adhesion (GO:0034112, FDR = 0.00622229238639), and the regulation of homotypic cell–cell adhesion (GO:0034110, FDR = 0.00623191022401).

The results of KEGG analysis are depicted in Appendix A. The first cluster of DEGs was significantly enriched in the 1 KEGG pathway (FDR < 0.05) for the peroxisome proliferator-activated receptor (PPAR) signaling pathway (FDR = 0.0031371681695). The second cluster of DEGs was significantly enriched in four KEGG pathways (FDR < 0.05) for valine, leucine, and isoleucine biosynthesis (FDR = 0.00291058069766), cysteine and methionine metabolism (FDR = 0.017404889651), the PPAR signaling pathway (FDR = 0.0244734061609), and arginine biosynthesis (FDR = 0.0303629614473). The third cluster of DEGs was significantly enriched (FDR < 0.05) in five KEGG pathways, cell cycle (FDR = 0.0000948931213108), progesterone-mediated oocyte maturation (FDR = 0.000783870051996), cellular senescence (FDR = 0.0152785111678), viral carcinogenesis (FDR = 0.0291390495005), and p53 signaling pathway (FDR = 0.0351285128146). The fourth cluster of DEGs was significantly enriched in 24 KEGG pathways (FDR < 0.05). Among all significantly enriched pathways, the top five most-enriched pathways included hematopoietic cell lineage (FDR = 0.000122365043445), primary immunodeficiency (FDR = 0.00023329041181), Th1 and Th2 cell differentiation (FDR = 0.000689800028747), toxoplasmosis (FDR = 0.00399123135862), and intestinal immune network for IgA production (FDR = 0.00452701834821).

Notably, in the first cluster of DEGs in the KEGG pathways, endocrine correlation was recorded. In the second cluster of DEGs in the KEGG pathways, three pathways were associated with amino acid metabolism and another one with endocrine relevance. In the third cluster of DEGs in the KEGG pathways, three pathways were associated with cell growth and death, one with the endocrine system, and one with cancer. Among the fourth cluster of DEGs in the KEGG pathways, 14 were associated with immunity and disease, four with amino acid and lipid metabolism, five with signal transduction, and one with the digestive system. These results indicated differences in the transcript levels in the liver of goat kids. DEGs of the first cluster, as well as the second cluster, were enriched to the same endocrine-related pathway. However, the Q values were different, and the upregulated genes were not identical in both comparisons.

### 3.4. Screening of the Key Regulatory Genes

To further identify the key regulatory genes that contributed to differential enrichment in the early developmental, metabolic, and immune-related pathways in the goat kid liver, PPI network analysis was performed (Figure 3). Based on the KEGG pathway enrichment outcomes, these four classes of DEG were significantly enriched in 34 pathways containing 148 genes.

All 148 genes were used for the PPI network construction, and 690 interactions between 148 genes were obtained. We sorted all genes in the network according to the degree of interaction and filtered out the 14 most important genes as potential key regulatory genes: *CD4*, *MYC*, *CCND1*, *ESR1*, *ZAP70*, *CD5*, *CDK1*, *IL7R*, *PLK1*, *AURKA*, *CCNA2*, *CD274*, *PPARG*, and *SELL*. According to the KEGG enrichment analysis, 38 pathways were significantly enriched, of which three pathways were associated with cell growth and death, four with the endocrine system, 24 with immunity and cancer, and seven with signal transduction. In addition, 15 of these 14 key genes were significantly enriched pathways overlapped with DEG significantly enriched pathways, namely cell cycle (FDR = 0.0000184219185935), progesterone-mediated oocyte maturation (FDR = 0.0000184219185935), primary immunodeficiency (FDR = 0.00023630691446), cellular senescence (FDR = 0.000672336234843), hematopoietic cell lineage (FDR = 0.00226613151582), breast cancer (FDR = 0.00372061978602), cell adhesion molecules (CAMs) (FDR = 0.00478423028459), viral carcinogenesis (FDR = 0.00883270624401), the pathways in cancer (FDR = 0.00950852829045), Epstein–Barr virus infection (FDR = 0.0103220971205), the p53 signaling pathway (FDR = 0.0108061020999), Th1 and Th2 cell differentiations (FDR = 0.0143991514955), T-cell receptor signaling pathway (FDR = 0.017736815761), Th17 cell differentiation (FDR = 0.0178677481378), and the PI3K-Akt signaling pathway (FDR = 0.0185674198687).

### 3.5. Validation of DEGs by qRT-PCR 

To validate the accuracy of the sequencing and analysis outcomes, we selected differently expressed mRNAs to validate the qRT-PCR results (Figure 4). qRT-PCR validation results were found to be roughly consistent with the RNA-seq results.

## 4. Discussion

Liver development involves significant changes in gene expression, which is mediated by transcriptional and post-transcriptional control [37]. This organ is an important hub for several physiological processes, including macronutrient metabolism, regulation of immune system support, blood volume, endocrine control of growth signaling pathways, lipid and cholesterol homeostasis, and the breakdown of exogenous compounds, such as drugs [38]. 

During early postnatal development in male rats, decreased β2-adrenergic receptor-mediated glucose mobilization may lead to lowered transcription of the receptor-encoding genes in the liver [39]. It is reported that liver growth factor promotes neurogenesis and neuronal survival, migration of new neurons, and striatal growth. The transcription factors *HNF3*, *MAZ*, and *Sp1* are required for elevated levels of S-protein gene expression in hepatocytes [40]. Research shows that liver X receptor α induces monounsaturated fatty acid synthesis in goat epithelial cells via the Srebp-1-dependent control of stearoyl-CoA desaturase 1 [41]. As is reported, antioxidant protection may play a role in chicken embryonic development, and that functional changes in postnatal liver development, correlating with chicken liver maturation [42].

Presently, various studies have focused on the metabolism and cellular differentiation of the liver. Studies on the transcriptome of the animal liver based on different developmental periods, as well as dietary structure, have been reported [43,44,45,46]. To understand the signaling pathways and key regulatory genes of the goat kids’ liver and functions related to immune regulation, this study used high-throughput transcriptome sequencing to detect the gene expression levels in the liver bodies of the goat kids at five different times points. More than 93% of Q30 was achieved in 25 samples. Compared with the reference genome, all genes identified in the library could be assigned to the reference genome of the goat, with the lowest assignment rate of 96.94%. In previous mammary gland and goatskin studies, the mapping did not exceed 80%, which indicates the high sequencing quality of our study [47].

To investigate the function of DE mRNAs with different expression profiles, we analyzed the DE mRNA expression patterns and identified four clusters. The DE mRNA expression profiles were related to the developmental stages in goats. We found that four clusters of target genes were enriched in the KEGG signaling pathway. DE mRNA was expressed at low levels at birth and at high levels in the liver of weaned goats (Cluster 1). In addition, they were significantly enriched in the PPAR signaling pathway, which is a widely studied nuclear receptor that regulates and controls metabolic changes in humans and animals. The PPAR is a nuclear receptor that mediates nutrient-dependent transcriptional activation and regulates the metabolic networks through energy homeostasis. Through different mechanisms, nutrients can regulate PPAR, thereby ultimately contributing to the prevention of various metabolic disorders [48]. Peroxisome proliferator-activated receptors include *PPAR-α*, *PPAR-β/δ*, and *PPAR-γ*, which play important roles in glucose and lipid metabolism [49]. Studies have shown that at knockout of *PPAR**-γ*, there is an interaction between glucose and lipid metabolism in adipose tissue, muscle, and liver, which indicates that *PPAR**-γ* plays an important role in maintaining systemic glucose and lipid homeostasis [50]. Mammalian milk is usually rich in lipids, and fatty acid metabolism is an important energy source for neonates in the early postpartum period [13,18]. The liver regulates lipid metabolism by associating with adipose tissue, while related transcription factors may contribute to fatty acid synthesis through the transfer of fatty acid transport protein (*FATP*) and fatty acid-binding protein (*FABP*) [38]. The liver is able to use fatty acids as an internal source of energy to other organs via an oxidative pathway, essential for organisms that experience extreme fasting or consume very low levels of dietary carbohydrates [51]. In our study, *FATP*, *FABP*, *PPARG*, and fatty acid transport-related genes upregulated, and PPARG is the key gene. We accordingly hypothesized that *FATP* and *FABP* act on fatty acid transport during lipid metabolism by activating *PPARG* through transport-related transcription factors acting on adipocytes, contribute to the prevention of various metabolic disorders, and provide energy. We recorded high expression of DE mRNA in the W2 and W8 groups for DEGs and low expression in the other groups (Cluster 2). Notably, the second group of DEGs was also significantly enriched in the PPAR signaling pathway. In our study, genes enriched downstream of the PPAR signaling pathway were also involved in cholesterol metabolism. In addition to lipid homeostasis, the liver is critical for the homeostasis of cholesterol in the body. Excess cholesterol from the diet and newly synthesized cholesterol is known to lead to inappropriate cell membrane dynamics [51]. We speculate that the PPAR signaling pathway may be activated by cholesterol in the diet during the mid-lactation and mixed milk feeding phase or by self-synthesized cholesterol. DE mRNA was highly expressed in the D1 and W4 groups and poorly expressed in the other groups (Cluster 3). The third group of DEGs was significantly enriched in five KEGG pathways (FDR < 0.05), namely cell cycle, progesterone-mediated oocyte maturation, cellular senescence, viral carcinogenesis, and the p53 signaling pathway. Most of these pathways were associated with cell growth and development. *CCNB1*, *CDK1*, and *RRM2* have been reported to be enriched in the p53 signaling pathway and are believed to act as potential biomarkers and therapeutic targets for hepatitis B virus-associated hepatocellular carcinoma [52]. *CDK1* was among the significantly enriched genes in the p53 signaling pathway in our study, and *CDK1* is also a key regulatory gene. At the heart of the cell cycle, CDKs propel cells through the various phases of the cell cycle. In the G1 phase, cyclin D-CDK4/6 (early) and cyclin E-CDK2 (late) prepare to enter the S-phase, at which point cyclin A-cdk2 takes over and coordinates replication, followed by activation of cyclin A/B-CDK1 to facilitate the transition from the G2 to the M phase [53]. *CDK1* can replace other CDKs, and it has been demonstrated to be sufficient to power the mammalian cell cycle [54]. *CDK1* functions outside mitosis as a general translational activator. It allows protein synthesis to directly adapt to the rate of cell proliferation [55]. We hypothesized that the goat kids liver cells grow rapidly at 2 and 8 weeks of age. There were several DEGs between the W12 and other groups, and their expression levels increased with age for the Laiwu black goats (Cluster 4). Among all of the significantly enriched pathways of Cluster 4 genes, most of them are related to immunity, and the top five most-enriched pathways were hematopoietic cell lineage, primary immunodeficiency, Th1 and Th2 cell differentiation, toxoplasmosis, and intestinal immune network. Among them, *CD4* gene was identified as the top one potential key regulatory gene of PPI network. CD4+ T cells are essential for achieving an effective immune response to pathogens, and they are activated after interacting with antigen MHC complex and differentiate into Th1 and Th2 subtypes, mainly according to the cytokine environment in the microenvironment [52]. The balance between Th1 and Th2 cells is important for normal immune response, and Th2 cell differentiation is usually promoted by Th1 response to maintain immune balance [56]. In our study, *CD4* genes were up-regulated with the increase of goat kid ages. We found that *CD4* genes were significantly enriched in the Th1 and Th2 cell differentiation signaling pathways and that the PPI regulatory network was a key regulatory gene. As an important immune organ, the liver is involved in regulating T cell differentiation and proliferation, and the proportion of CD4+ T cells in the liver of adult animals is significantly higher than that in the peripheral blood [57,58]. The gradually increased expression of the *CD4* gene found in this study indicates that the amount of CD4+ T cells in goat liver and their differentiation are gradually activated and enhanced over time during the neonatal stage of goat kids. We accordingly hypothesized that *CD4* genes play an important role in the normal immune responses by acting in the Th1 and Th2 cell differentiation signaling pathways.

A total of 690 interactions among 148 genes were obtained in the PPI regulatory network constructed from the KEGG pathway enrichment results. A total of 14 key regulatory genes were screened in order of the degree of interaction: *CD4*, *MYC*, *CCND1*, *ESR1*, *ZAP70*, *CD5*, *CDK1*, *IL7R*, *PLK1*, *AURKA*, *CCNA2*, *CD274*, *PPARG*, and *SELL*. *MYC* has been found to be unregulated in almost half of the human solid tumors and leukemias, and increased levels of *MYC* has been attributed to retroviral transfection or transduction of circulating mammalian cells that reduces the need for growth factors, prevents cell cycle secretion, accelerates cell distribution, and increases the number of cells [59]. The oncoprotein *MYC* is an important transcription factor that regulates the growth, proliferation, and expression of genes involved in different cellular signaling pathways [60]. *MYC* dysregulation has been causally linked to the development, persistence, and progression of cancer, although possible mechanisms by which oncoproteins play a role in these processes include increased cell proliferation, inhibition of cell death, metabolic regulation, promotion of angiogenesis, and regulation of stem cell formation [61,62,63]. We thus speculated that *MYC* genes play an important role in early development-related mechanisms in the liver of goat kids. Thus, we believe that the functions of these key genes deserve further investigation.

## 5. Conclusions

In this study, the transcriptome of the goat livers at five developmental stages during the juvenile age were sequenced, and DEGs functional enrichment analysis showed a major concentration in the pathways related to early development, substance metabolism, and immunity. This study provides important information for facilitating the study of the role of the liver in the early development, metabolism, and immunity at a young age. Our results extend the current goat mRNA database and contribute to the better understanding of the mechanisms of early development from the perspective of ruminant liver development.

## Figures and Tables

**Figure 1 genes-13-00833-f001:**
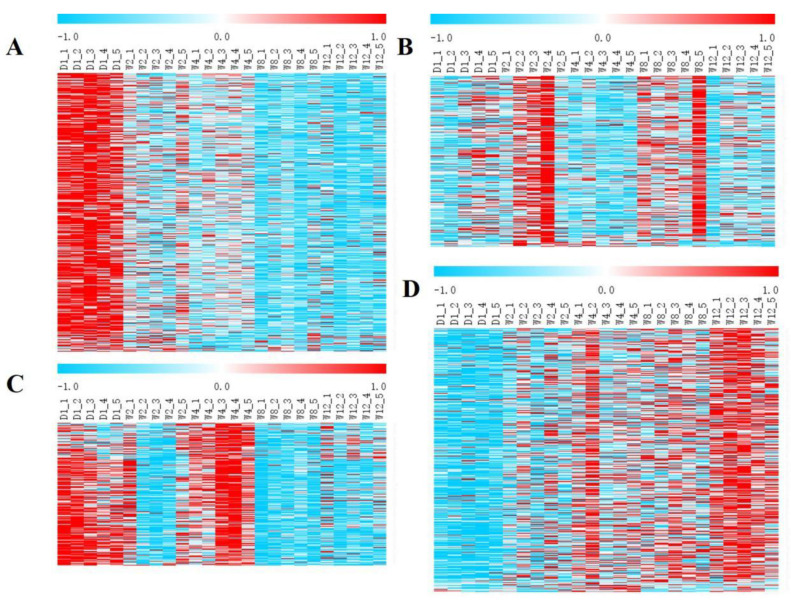
Clustering pattern of DEGs between different groups. (**A**) Cluster 1 of DEGs between different groups. The DEGs expression levels decreased with age in Laiwu black goats. (**B**) Cluster 2 of DEGs between different groups. A high expression of DEGs in the W2 and W8 groups and a low expression in the other groups. (**C**) Cluster 3 of DEGs between different groups. A high expression of DEGs in the D1 and W4 groups and an upregulation of the low expression in the other groups. (**D**) Cluster 4 of DEGs between different groups. The DEGs expression levels increased with age.

**Figure 2 genes-13-00833-f002:**
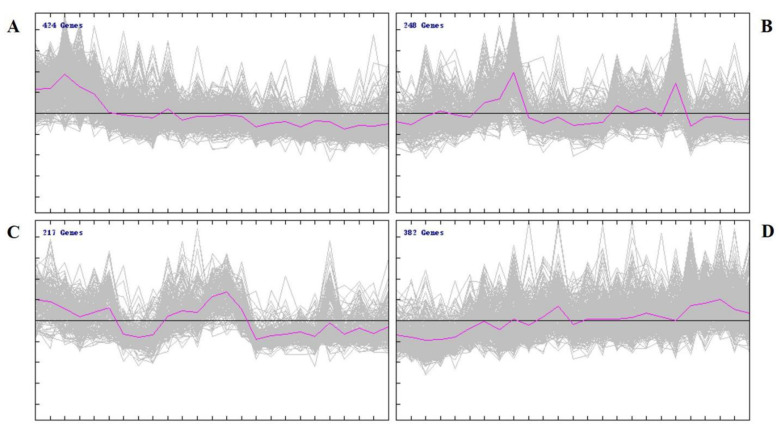
Expression trends of DEGs (**A**) Cluster 1 of DEGs between different groups. The DEGs expression levels decreased with age in the Laiwu black goat. (**B**) Cluster 2 of DEGs between different groups. A high expression of DEGs in the W2 and W8 groups and a low expression in the other groups. (**C**) Cluster 3 of DEGs between different groups. A high expression of DEG in the D1 and W4 groups and an upregulation of the low expression in the other groups. (**D**) Cluster 4 of DEGs between different groups. The DEGs expression levels increased with age.

**Figure 3 genes-13-00833-f003:**
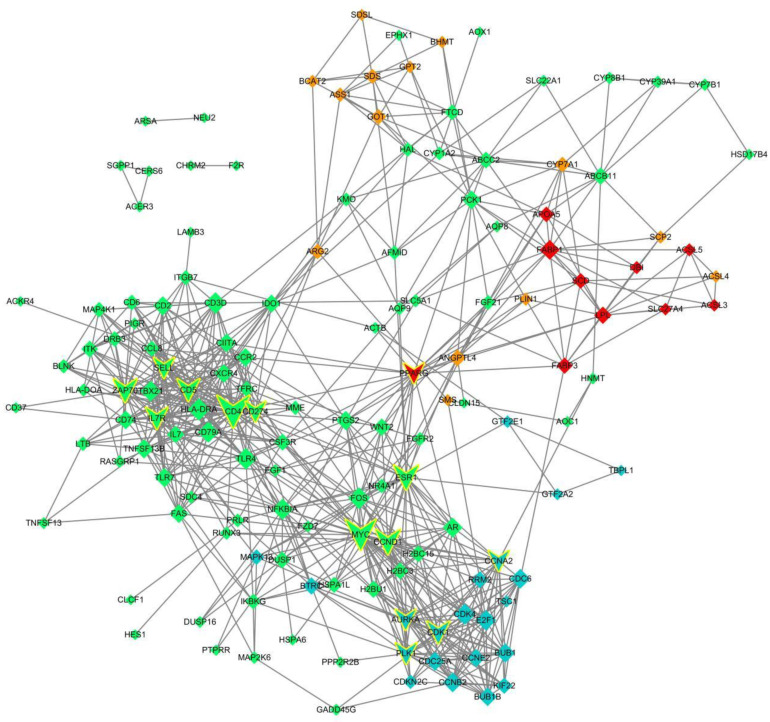
Protein–protein interaction networks of the KEGG pathway significantly enriched genes. Red indicates genes in cluster 1; orange indicates genes in cluster 2; blue indicates genes in cluster 3; green indicates genes associated with cluster 4; those with yellow boxes indicate key genes.

**Figure 4 genes-13-00833-f004:**
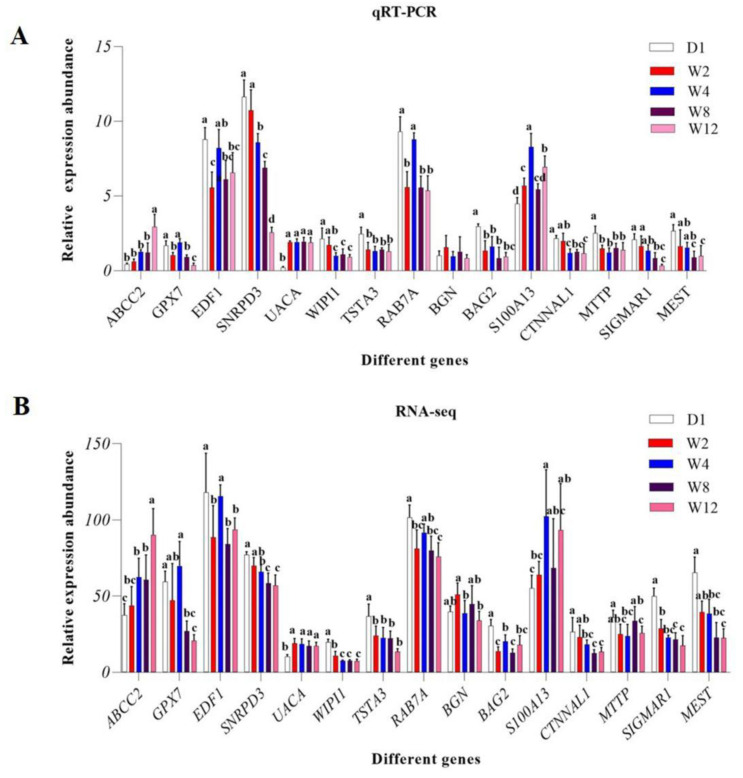
Comparison of relative expression of DEGs between RNA-Seq and qRT-PCR results. (**A**) The results of qRT-PCR. (**B**) The results of RNA-seq. Significant differences are indicated by different symbols with FDR < 0.05 in RNA-Seq results and *p* < 0.05 in qRT-PCR results.

**Table 1 genes-13-00833-t001:** Primer sequences for qRT-PCR testing of randomly selected DEGs.

Gene Name	Up	Down
*ACTB*	CACCACACCTTCTACAAC	TCTGGGTCATCTTCTCAC
*ABCC2*	AATACAACGGCACCAACTATCCATCC	CATAGACACTCCAGATGTTCGCTACG
*GPX7*	GGAGCAGGACTTCTACGACTTCAAG	CCACATTCACCACCAGGGACAC
*EDF1*	GTGGTGGATGAAAATGCAGTAG	TTGAAAATCCACAGCCAATGAG
*SNRPD3*	GTATGACAGGCTATGACCCGTTGC	CGTTCTGGAAGATCAGCGACACTC
*UACA*	GACACCGCTTGTTCTGGCTACTC	GCAACCGTACTCGCATCCTAGC
*WIPI1*	CAGTTCAGTCGCTCAGTCGTGTC	GGAAACTGTTGGTTGTACAGAC
*TSTA3*	GCTACTCCGTGTATGTGTACAA	TAACGAAGGAATTCATGATGCC
*RAB7A*	CAGGAGGCAACACGGTGAAGAC	CACAGACAGACGAGATGGCTTGATG
*BGN*	GTAAACCACCTGCACCTAAAAG	CCTCCACAGTACACAGTACAAT
*BAG2*	CTTTGAGAGAAGCAGCAACTG	TGACACTTCAACGGTGAGAG
*S100A13*	TGTGCTTAGTTGCTCAGTCGTGTC	AGGTTGGATCTCTGGGTTGGGAAG
*CTNNAL1*	TGGGAGATGGTGAAGGACAGAGAAG	TTGTGGTTGTTCAGTCGCTCAGTC
*MTTP*	CAGTTCAGTCGCTCAGTCGTGTC	CGAAAGAGGATGAGATGGCTGGATG
*SIGMAR1*	ATGCCCTCCTTTCTGCCGTTTG	GAAGGTGCCTGAGATGATGGTATCG
*MEST*	ATACCCGACCTTCTGAGAGTGAGC	CCCACCCAGCGTCTTCTAAACTTC

**Table 2 genes-13-00833-t002:** Statistics of known and novel transcripts in each group.

Sample Name	Known mRNA Num	Novel mRNA Num	All mRNA Num
D1	23,861 (73.37%)	7324	31,185
W2	24,022 (76.58%)	7347	31,369
W4	23,901 (76.57%)	7315	31,216
W8	23,725 (76.37%)	7339	31,064
W12	23,799 (76.42%)	7344	31,143
Total	29,976 (79.94%)	7521	37,497

## Data Availability

The original contributions presented in the study are publicly available. All data can be downloaded from the Gene Expression Omnibus (GEO) database, and the accession number is: GSE194190.

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
