# Peer review of "High-Throughput Sequencing Reveals Transcriptome Signature of Early Liver Development in Goat Kids"

_genes, 2022, doi:10.3390/genes13050833_

Round 1

Reviewer 1 Report

The introduction part is too general, with basic information. The reviewer would expect deeper inside in current stage of knowledge about kid liver.

line 67, please use "sacrificed" instead of "slaughtered" - would be good to describe the procedure, to proof it was humanitary

line 69 "time point"

Any health test checks of kid goats? The goats were kept individually? Any histological analysis of liver samples? Please add some info about the goat breed. Which parturition?

Figures legends are very brief - please add important information

the nutrients change hypothesized in discussion part is not confirmed by the nutrients level study.

line 354 - please explain the theory with hepatocellular carcinoma cells in goat kids

The Reviewer would expect in the discussion part deeper analysis of the differences in the liver between the chosen time points& way of nutrition

Author Response

Dear reviewer,

Thank you for your very valuable comments, we found them very reasonable and helpful. Our detailed modification information were attached as a word file, please check.

Reviewer 2 Report

The manuscript “High-throughput sequencing reveals transcriptome signature of early liver development in goat kids” is of interest and per the scope of the journal. It is very well structured and complete, they perform a transcriptional analysis of goat liver samples at different periods of the first weeks after birth in goat kids to examine the differentially expressed genes. A few corrections need to be done before being published. 

In general, the vocabulary used is not in accordance with the species, for example, “breastfeeding” is a term used for humans, in goats, it would be better to use “nursing” or “suckled from their mother”, also in L64 it says “goat children”, could be “goat kids”.

Also, genes’ names should be in italics. 

Abbreviations should be defined at their first mention. 

Consider placing the Table S1 information about primers as an actual table.

Author Response

(The authors gave the same response as above.)
